# Clustered rapid induction of apoptosis limits ZIKV and DENV-2 proliferation in the midguts of *Aedes aegypti*

Jasmine B. Ayers [1,2,3], Heather G. Coatsworth [1,3], Seokyoung Kang[1,3], Rhoel R. Dinglasan [1,3 ✉] & Lei Zhou [1,2 ✉]

Inter-host transmission of pathogenic arboviruses such as dengue virus (DENV) and Zika virus (ZIKV) requires systemic infection of the mosquito vector. Successful systemic infection requires initial viral entry and proliferation in the midgut cells of the mosquito followed by dissemination to secondary tissues and eventual entry into salivary glands[1]. Lack of arbovirus proliferation in midgut cells has been observed in several *Aedes aegypti* strains[2], but the midgut antiviral responses underlying this phenomenon are not yet fully understood. We report here that there is a rapid induction of apoptosis (RIA) in the *Aedes aegypti* midgut epithelium within 2 hours of infection with DENV-2 or ZIKV in both in vivo blood-feeding and ex vivo midgut infection models. Inhibition of RIA led to increased virus proliferation in the midgut, implicating RIA as an innate immune mechanism mediating midgut infection in this mosquito vector.

---

[1] Emerging Pathogens Institute, University of Florida, Gainesville FL 32611, USA. [2] Department of Molecular Genetics & Microbiology, College of Medicine, University of Florida, Gainesville FL 32603, USA. [3] Department of Infectious Diseases & Immunology, College of Veterinary Medicine, University of Florida, Gainesville FL 32608, USA. ✉email: rdinglasan@epi.ufl.edu; leizhou@ufl.edu

ZIKV and DENV serotypes 1–4 are arthropod-borne viruses (arboviruses) of the genus *Flavivirus* that cause acute febrile illness in humans. Upon reinfection with a different dengue viral serotype, antibody-dependent enhancement can result in severe and potentially fatal manifestations such as hemorrhagic fever and dengue shock syndrome[3]. There are an estimated 100 million symptomatic dengue infections annually[4]. Although acute infection with ZIKV is typically asymptomatic or mild, it has been found to cause neurological sequelae such as Guillain–Barre syndrome in adults and neurological birth defects such as microcephaly in infants born to infected mothers[5,6].

DENV and ZIKV are both primarily transmitted between humans by *Aedes aegypti*. The primary site of viral infection is the mosquito midgut epithelium, and the midgut is the first physical defense barrier against virus establishment. The proliferation of arboviruses in the midgut following consumption of an infected blood meal is a prerequisite for subsequent systemic infection of the vector and eventual transmission of the virus. Interestingly, not all *A. aegypti* have an equivalent ability to contract and transmit DENV and ZIKV[7]. Failure to establish viral infection in the midgut after a flavivirus-infected blood meal has been observed in both field-derived lab *A. aegypti* strains such as the Cali-Midgut Infection Barrier strain and in wild-collected individuals[2,8–10]. The mechanisms by which mosquitoes resist these pathogens are of great epidemiological interest, both for predicting the ability of local mosquito populations to transmit these viruses and for engineering mosquito strains that could resist infection.

Apoptosis has been posited as an important innate immune response against viral infection in both insects and mammals[11–13]. One line of evidence is that viral genes with antiapoptotic function are crucial for infectivity of insects by several families of virus[14,15]. ZIKV in particular has recently been shown to encode subgenomic RNAs implicated in inhibiting apoptosis and increasing infectivity in mosquitoes[16]. In addition, expression of pro-apoptotic regulatory genes by genetically engineered Sindbis virus demonstrated that inducing apoptosis can limit viral infection of *A. aegypti*[17]. However, depending on the virus/vector pairing studied, apoptosis observed following arbovirus infection of mosquitoes correlates in some cases with refractoriness[16–21] and in other cases with susceptibility[22–24]. Apoptosis in insects is in some cases a mechanism that facilitates

viral release- in Rift Valley Fever virus infection of *Anopheles stephensi*, apoptosis appears to be necessary for viral escape into the salivary gland lumen[23]; the baculovirus *Autographa californica* M nucleopolyhedrovirus requires caspase activation to pass through the midgut basal lamina[25], and *Spodoptera frugiperda* ascovirus encodes its own executioner caspase, which is expressed during late stages of the replication cycle[26]. The seemingly contradictory roles of apoptosis in insect virus infection suggest a balancing point between infection control and pathological tissue damage. The kinetics of the apoptotic response in relation to the viral life cycle appears to be an important factor in that balance. Previous work on apoptosis following viral infection in the adult mosquito midgut has focused on 24+ hours post infection (hpi)[16–24]. However, experiments conducted in *Drosophila* indicate that a apoptosis, which occurs within 4 hours following injection of flock house virus is responsible for limiting systemic infection[27]. We have termed the appearance of apoptotic cells before 4 hpi as a rapid induction of apoptosis (RIA) to distinguish the phenomenon from the much larger, nonspecific induction of apoptosis we observed at later timepoints post-feeding in hematophagous *A. aegypti*. The present study explores whether this RIA response and its limiting effect occurs in vector mosquitoes infected with medically relevant *Flaviviruses*.

## Results and discussion

Using adult female Orlando (ORL) strain *A. aegypti*, we assayed for signs of apoptosis following exposure to a naive blood meal or a blood meal containing either DENV-2 NGC (DENV serotype 2 New Guinea C strain) or ZIKV-PR (lab-adapted Puerto Rico ZIKV strain PRVABC59). At 2 hours after blood-feeding began, the midguts were dissected and immediately fixed (Fig. 1a). Using terminal deoxynucleotidyl transferase dUTP nick-end labeling (TUNEL) to detect the DNA fragmentation, we noted only a few apoptotic cells in the midgut epithelium of mosquitoes fed with a naive blood meal (Fig. 1b). In contrast, there was a 29.1-fold increase ($p = 3.74e\text{-}06$) in mean TUNEL-positive cells in the midgut of mosquitoes that consumed a blood meal containing ZIKV (Fig. 1c), and a 19.6-fold increase in DENV-2 fed mosquitoes ($p = 1.18e\text{-}06$) (Fig. 1d). The few TUNEL-positive cells in naive blood-fed midguts were discrete and surrounded by cells that were negative for TUNEL (Fig. 1b). In contrast, in DENV-2

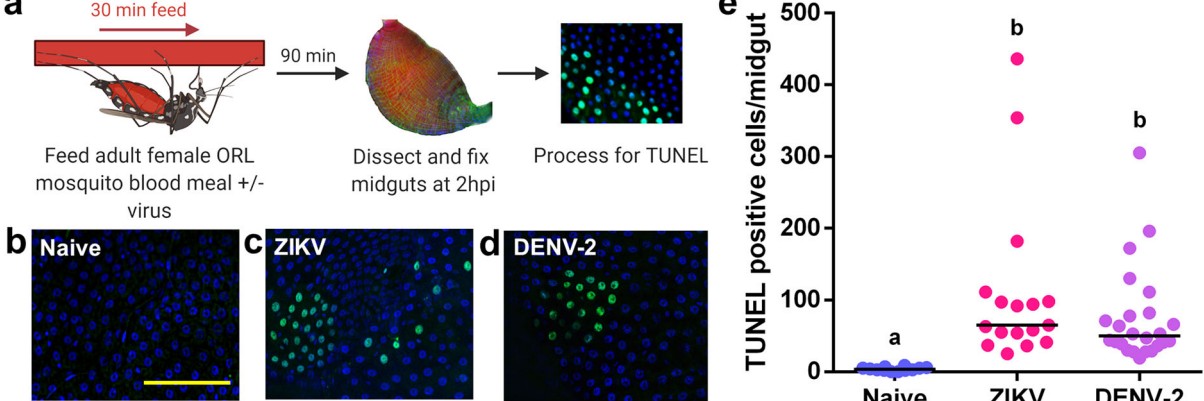

**Fig. 1 Exposure to a blood meal containing DENV-2 or ZIKV induces DNA fragmentation in the midgut epithelium of adult female *Aedes aegypti* (ORL) mosquitoes at 2hpi. a** Workflow diagram of in vivo infection. **b–d** Representative images of DNA fragmentation visualized via TUNEL with DAPI counterstain in midguts from adult female ORL mosquitoes at 2 hours post **b** naive blood meal **c** ZIKV infected blood meal or **d** or DENV-2-infected blood meal. Scale bar shown is 100 μm in length and can be applied to all images. **e** Quantification of TUNEL-positive cells per midgut at 2 hours post in vivo infection ($n$ (naive) = 12; $n$ (ZIKV) = 17; $n$ (DENV-2) = 24). The horizontal line indicates the median. Treatments without a common letter were found to be statistically significant ($\alpha = 0.05$) as calculated by Kruskal–Wallis test with Mann–Whitney post hoc comparison (Kruskal–Wallis chi-squared = 32.017, $df = 2$, $p$ value = 1.116e-07).

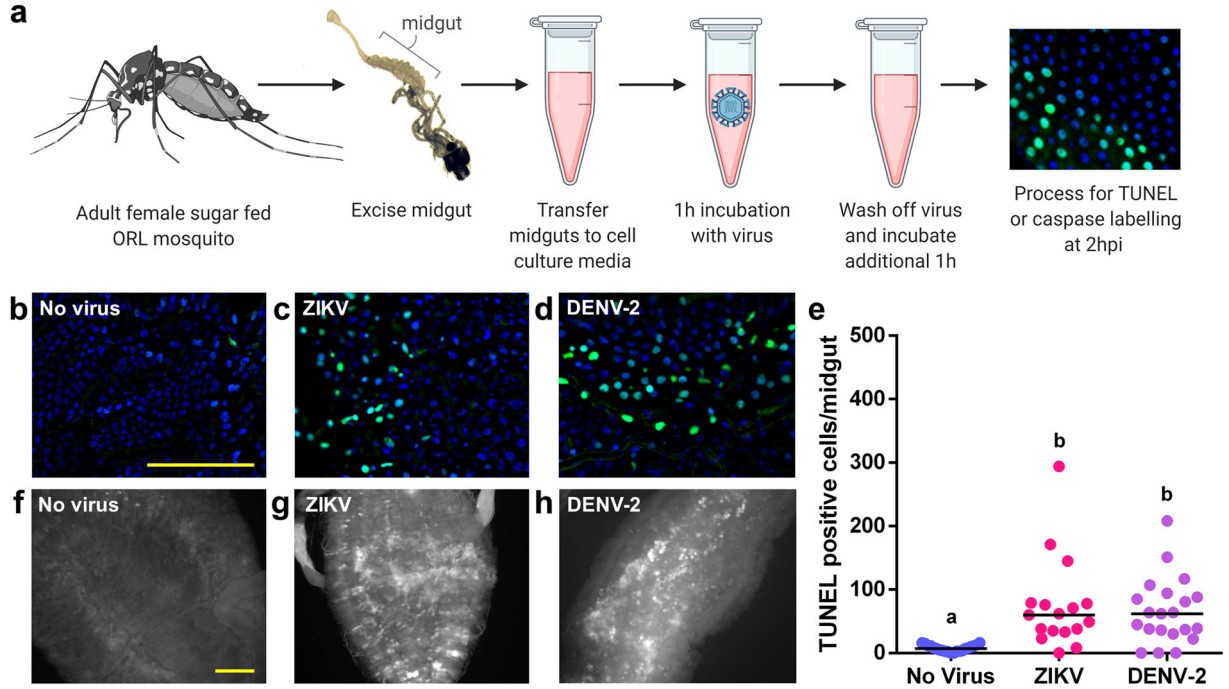

**Fig. 2 Exposure to a blood meal containing DENV-2 or ZIKV induces DNA fragmentation in the epithelium of ex vivo adult female *Aedes aegypti* (ORL) midguts at 2hpi. a** Workflow diagram of ex vivo midgut infection. **b–d** Representative images of DNA fragmentation visualized via TUNEL at 2 hours after **b** ex vivo mock infection, **c** ZIKV infection or **d** DENV-2 infection. **e** Quantification of TUNEL-positive cells per midgut at 2 hours post ex vivo infection. ($n$ (naive) = 21; $n$ (ZIKV) = 17; $n$ (DENV-2) = 18). The horizontal line indicates the median. Treatments without a common letter were found to be statistically significant ($\alpha = 0.05$) as calculated by Kruskal–Wallis test with Mann–Whitney post hoc comparison (Kruskal–Wallis chi-squared = 22.285, df = 2, p value = 1.448e-05). **f–h** Pan-caspase activation at 2 hours after **f** mock infection **g** ZIKV infection or **h** DENV-2 infection. Scale bars shown are 100 μm in length and can be applied to all related images.

or ZIKV-fed midguts, TUNEL-positive cells often appeared in clusters of 10–50 cells (Figs. 1c, 1d). The position of the TUNEL-positive cells in the midgut following virus feeding appears to be random and not restricted to particular regions of the midgut. An increase in TUNEL-positive cells in in vivo ORL midguts can be detected as soon as 1 hour following a ZIKV blood meal, and this elevation persists to 4 hpi (Supplementary Fig. 1). At 8 hpi and 24 hpi, both naive blood-fed mosquitoes and virus-fed mosquitoes had an equivalent, high number of TUNEL-positive cells (mean of 465 TUNEL-positive cells/gut; Supplementary Fig. 1), which is consistent with previous work showing a high level of midgut turnover in hematophagous mosquitoes during blood meal digestion[28].

To orthogonally verify that exposure to ZIKV or DENV-2 induces RIA in the midgut, we developed an ex vivo infection model, whereby midguts were dissected from adult female ORL mosquitoes that had not been blood fed and were incubated at room temperature in either media alone or media containing virus ($10^6$ PFU/mL) (Fig. 2a). At 2 hours following exposure to either ZIKV or DENV-2, there was a 9.8-fold increase in mean TUNEL-positive cells per midgut over the no virus control in ZIKV-treated midguts ($p = 3.59e-05$) and a 9.1-fold increase in DENV-2-treated midguts ($p = 0.000433$) (Fig. 2b–e). The patterning of these TUNEL-positive cells was similar to what was seen in the in vivo infection model, as we frequently observed clusters of 10–50 cells with no obvious anterior to posterior preference in localization. In addition to DNA fragmentation, there was an extensive activation of caspases detected via a pan-caspase in situ activation reporter in the midgut 2 hours after DENV-2 or ZIKV exposure (Fig. 2f–h).

To test whether the RIA response correlated with a refractory phenotype in strains with differential virus susceptibility, we chose the MOYO-Refractory (MOYO-R) and MOYO-Susceptible (MOYO-S) strains of *A. aegypti*, which were established with selective inbreeding from the original MOYO-In-Dry strain for their respective resistance and susceptibility to *Plasmodium gallinaceum* (avian malaria parasite)[29]; they were later found to be refractory (19.54% infection rate) and susceptible (56.60% infection rate), respectively, to DENV-2 (Jamaica 1409 strain) infection[30]. When both MOYO-R and MOYO-S mosquitoes were fed with a DENV-2-infected blood meal, there was a 1.8-fold higher mean number of TUNEL-positive epithelial cells in MOYO-R mosquitoes as compared to MOYO-S mosquitoes ($p = 0.00212$) (Fig. 3).

We hoped to use Z-Vad-FMK or another small molecule caspase inhibitor to confirm the role of caspase activation in the RIA phenotype observed by TUNEL. However, addition of dimethyl sulfoxide solvent alone in the ex vivo infection media reduced the appearance of TUNEL-positive cells after virus infection (Supplementary Fig. 2).

To see the effect of a lack of RIA on the course of infection, we used human Alpha-1 anti-trypsin (hAAT) as a water-soluble apoptosis inhibitor. hAAT is a serum protein serine protease, which has also been found to inhibit caspase-3 activation and subsequently suppress apoptosis in human cells[31]. hAAT is also involved in the human acute infection response as up to a four-fold increase in hAAT serum concentration was found following stimulation of innate immune cells in donor blood with heat killed *Staphylococcus epidermidis*[32], although induction of hAAT has not been observed in human flavivirus infection[33].

We found that supplementation of the infective blood meal with 10 mg/mL of clinical grade hAAT (serum level of hAAT in healthy subjects is around 1.5–3.5 mg/mL) suppressed RIA following exposure to ZIKV or DENV-2, as evidenced by a

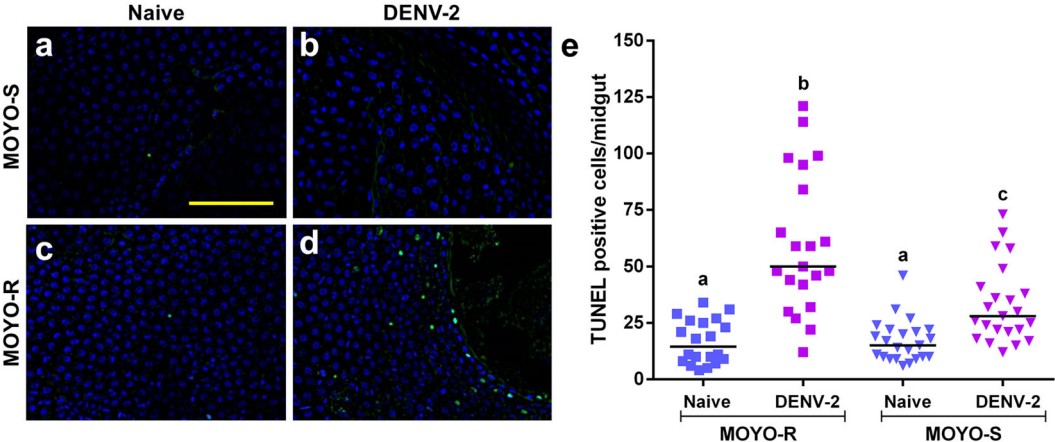

**Fig. 3 *Aedes aegypti* MOYO-R strain mosquitoes that are partially resistant to DENV-2 have higher rates of TUNEL-positive cells in the midgut than DENV-2 susceptible MOYO-S strain mosquitoes at 2 hours post ingestion of a DENV-2-infected blood meal. a–d** Representative images of TUNEL with DAPI counterstain in midguts from adult female mosquitoes at 2 hours post blood meal. **a** MOYO-S mosquitoes fed naïve blood, **b** MOYO-S mosquitoes fed DENV-2-infected blood meal, **c** MOYO-R mosquitoes fed naïve blood or **d** MOYO-R mosquitoes fed DENV-2-infected blood. Scale bar = 100 μm. **e** Quantification of TUNEL-positive cells per midgut at 2 hours post in vivo infection (*n* (MOYO-R naïve) = 20; *n* (MOYO-R DENV-2) = 21; *n* (MOYO-S naïve) = 23; *n* (MOYO-S DENV-2) = 24). The horizontal line indicates the median. Treatments without a common letter were found to be statistically significant (α = 0.05) as calculated by Kruskal–Wallis test with Mann–Whitney post hoc comparison (Kruskal–Wallis chi-squared = 41.907, df = 3, p value = 4.199e-09). Scale bar shown is 100 μm in length and can be applied to all images.

significant decrease in the number of TUNEL-positive cells in hAAT-supplemented mosquitoes (p = 4.62e-06, and 2.38e-05, respectively) (Fig. 4b, c). The suppression of apoptosis by hAAT supplementation is short-lived, with levels of TUNEL-positive cells in supplemented mosquitoes returning to control infection levels by 24 hpi (Supplementary Fig. 3). This implies that the effect of apoptosis inhibition is specifically owing to RIA at early infection timepoints.

Addition of hAAT also increased viral replication in this infection model. hAAT supplementation caused a significant reduction in mean difference in cycle threshold (ΔCT) from 13.59 to 6.47 in ZIKV infected mosquitoes (p = 1.74e-08). This reduction was less pronounced but still significant (12.59–10.03) in DENV-2-infected mosquitoes (p = 0.00681) (Fig. 4d, e). At 7 days post infection (dpi), significantly more mosquitoes in the hAAT-supplemented ZIKV group were virus positive in the midgut (91% vs 35% for ZIKV, p = 5.49e-11) (Fig. 4f). However, there was no significant difference in the number of virus positive mosquitoes in the DENV-2 hAAT-supplemented group (72% vs 54%, p = 0.0645) (Fig. 4f). Furthermore, infected hAAT-supplemented mosquitoes had higher titers of infectious virus in the midgut than non-supplemented mosquitoes, with a 15.1-fold increase in mean PFU/midgut in ZIKV infected mosquitoes (p = 1.10e-12) and a 6.8-fold increase in DENV-2 infected mosquitoes (p = 0.00193) (Fig. 4f). As susceptibility to infection could be increased by potential adverse side effects of hAAT supplementation, we also recorded mosquito mortality at the 7 dpi harvest timepoint. There was no observable correlation between mortality and hAAT supplementation (for ZIKV p = 0.183, for DENV-2 p = 0.637, by Chi-squared test) (Supplementary Table 1).

Whether the correlation between a virus-refractory phenotype and a strong RIA response holds true in a broader panel of strains is an interesting topic for future investigation. Several studies have observed increased expression of pro-apoptotic genes correlated with refractory phenotypes in *A. aegypti* strains[18,20,34]. A previous study reported induction of the pro-apoptotic gene *Michelob_x* (*mx*), detected by qRTPCR, at 3 hpi with DENV-2 (Jamaica 1409 strain) in the midgut of MOYO-R but not MOYO-S[27].

In order to begin investigating the mechanism by which virus exposure triggers apoptosis, we performed transcriptomic analysis by qRTPCR on three known upstream regulators of caspase activation in *A. aegypti*, which have previously been implicated in the innate immune response. Mx and IAP-antagonist Mx-like protein (IMP) are the characterized members of the inhibitor of apoptosis (IAP)-antagonist gene family in *A. aegypti*, which induce apoptosis by alleviating inhibition of caspases by IAPs[35]. In our ORL infection model, transcript level of *mx* was significantly increased in ZIKV-fed mosquitoes but not in DENV-2 fed mosquitoes. *IMP* transcript level was not impacted by infection status (Supplementary Fig. 4). Defense repressor 1 (DNR1) is a negative apoptosis regulator, which has been characterized in *A. aegypti*, *Aedes albopictus* cells and *Drosophila* as depleting levels of the caspase dronc by ubiquitination[18,36,37]. DNR1 is also a negative regulator of the immune deficiency pathway in *Drosophila*[38] and is depleted following Sindbis virus infection of *Aedes albopictus* cells[37], making it a gene of interest for investigating cross-talk between innate immunity and apoptosis. In this infection model, there was no change in transcript level of *DNR1* at the 2hpi timepoint between naïve blood and virus-fed mosquitoes (Supplementary Fig. 4). In the future, studies investigating protein-level regulation of these genes during the RIA response should be undertaken to confirm involvement or lack thereof.

Bioinformatic analyses[39] of the recently updated *A. aegypti* genome (version AaegL5)[40] indicated that additional IAP-antagonists are present besides Mx and IMP. The presence of these multiple IAP-antagonists could be imperative for the RIA response, as deleting the regulatory region responsible for coordinated induction of three IAP-antagonist genes (*reaper*, *sickle*, and *hid*) in *Drosophila* adults significantly suppressed RIA and rendered the flies more susceptible to flock house virus[27]. Whether an orthologous regulatory region in *A. aegypti* is present and responsible for mediating the RIA response to the flaviviruses tested herein remains to be seen. The differential RIA response observed in *A. aegypti* strains could be due to genetic polymorphisms present in enhancer or regulatory regions as polymorphisms in cis-regulatory elements between MOYO-R and MOYO-S have previously been observed[41].

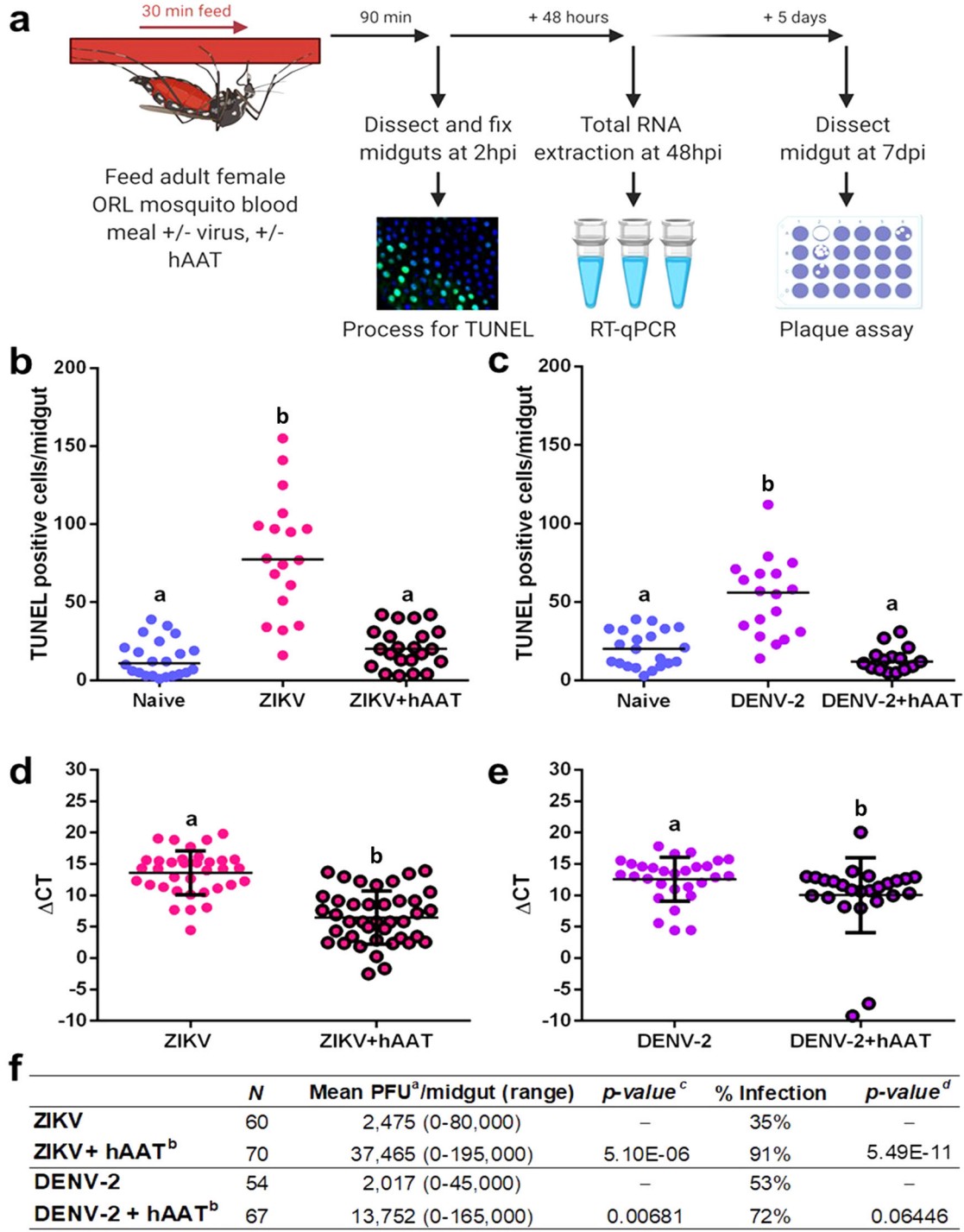

| | N | Mean PFU[a]/midgut (range) | p-value[c] | % Infection | p-value[d] |
|---|---|---|---|---|---|
| ZIKV | 60 | 2,475 (0-80,000) | – | 35% | – |
| ZIKV + hAAT[b] | 70 | 37,465 (0-195,000) | 5.10E-06 | 91% | 5.49E-11 |
| DENV-2 | 54 | 2,017 (0-45,000) | – | 53% | – |
| DENV-2 + hAAT[b] | 67 | 13,752 (0-165,000) | 0.00681 | 72% | 0.06446 |

[a]Plaque-forming units;  [b]Concentration = 10 mg/mL;  [c]Welch's Unpaired t-test;  [d]χ2 test

An interesting feature of the RIA in our infection models was the clustering of TUNEL-positive cells. There is no previous evidence suggesting a clustered sub-population of cells in the midgut epithelium would be particularly susceptible to DENV-2 or ZIKV infection. Therefore, while the presence of virus in any or all cells undergoing RIA (i.e., TUNEL-positive) remains to be seen, we speculate that cells adjacent to those infected may also undergo apoptosis (Fig. 5). In mammalian systems, virus infection-induced apoptosis of uninfected cells has been observed in Herpes Simplex Virus-1 and Human Immunodeficiency virus infection[42,43]. In plants, the rapid induction of cell death

following pathogen infection often involves cells surrounding the infected cells[44,45]. This type of cell death, termed hypersensitive response, is responsible for mediating resistance to pathogen infection in higher plants. The mechanism of hypersensitive response in plants has not been completely understood, but likely involves the production and release of reactive oxygen species[46,47]. Apoptosis-induced-apoptosis, in which a cell undergoing apoptosis stimulates apoptosis in its neighbors by paracrine signaling, has been observed in *Drosophila* and is mediated by Eiger, the insect ortholog of mammalian tumor necrosis factor[48]. An Eiger ortholog (AAEL010524-PA) has been identified in *A.*

**Fig. 4 Supplementing a DENV-2 or ZIKV blood meal with hAAT in adult female Aedes aegypti (ORL) mosquitoes inhibits the rapid induction of apoptosis in the midgut at 2hpi and increases subsequent midgut viral replication. a** Workflow of the apoptosis inhibitor co-feeding experiment, hpi=hours post infection, dpi=days post infection. **b–c** TUNEL-positive cell counts in midguts from mosquitoes fed a **b** ZIKV (n (naive) = 22; n (ZIKV) = 18; n (ZIKV + hAAT) = 22) or **c** DENV-2-infected blood meal or a virus-infected blood meal supplemented with 10 mg/mL hAAT (n (naive) = 21; n (DENV-2) = 18; n (DENV-2 + hAAT) = 15). Treatments without a common letter were found to be statistically significant ($\alpha = 0.05$) as calculated by Kruskal–Wallis test with Mann–Whitney post hoc test (panel a: Kruskal–Wallis chi-squared = 32.532, df = 2, p value = 8.624e-08; **b** Kruskal–Wallis chi-squared = 26.513, df = 2, p value = 1.749e-06) **d–e** ΔCT values from RT-qPCR detecting **d** ZIKV (n (ZIKV) = 34; n (ZIKV + hAAT) = 40) or **e** DENV-2 genome copy in whole mosquitoes 48 hours after an infected blood meal with or without 10 mg/mL hAAT (n (DENV-2) = 29; n (DENV-2 + hAAT) = 25). The horizontal line indicates the median. Treatments without a common letter were found to be statistically significant ($\alpha = 0.05$) as calculated by Mann–Whitney U test (**d** W = 905, p value = 1.735e-08; **e** W = 466, p value = 0.006808). **f** Productive midgut infection quantified by midgut plaque assay at 7 days after feeding on a ZIKV or DENV-2-infected blood meal (n (ZIKV) = 34; n (ZIKV + hAAT) = 40; n (DENV-2) = 29; n (DENV-2 + hAAT) = 25)). Treatments without a common letter were found to be statistically significant ($\alpha = 0.05$) as calculated by Mann–Whitney U test (ZIKV W = 608, p value = 1.102e-12; DENV-2 W = 1228.5, p value = 0.001929).

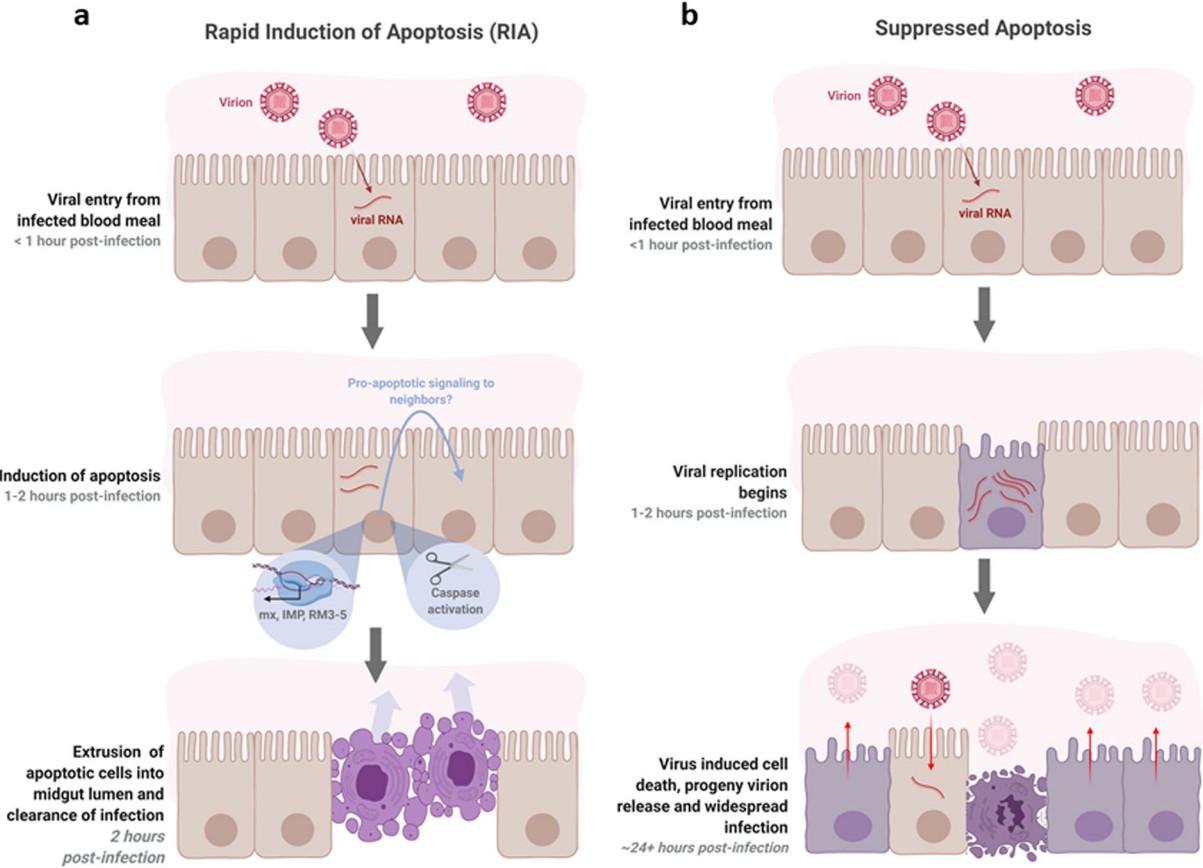

**Fig. 5 Hypothesized model of antiviral apoptosis in the *Aedes aegypti* midgut. a** RIA of infected cells in the midgut epithelium within 2 hours of an infected blood meal arrests viral replication before progeny virions can be produced and suppresses infection. The clustered appearance of apoptotic cells may be caused by pro-apoptotic signaling to neighboring cells. **b** At later timepoints post infection, when infection is widespread throughout the midgut and progeny virions are being produced, death of infected cells may facilitate release of virions from cells and/or from the midgut by degrading the integrity of the epithelium and basal lamina.

*aegypti.* Previous research on West Nile virus-infected *Culex quinquefasciatus* cells suggests that paracrine signaling from infected cells induces an antiviral state in their neighbors via activation of the Janus kinase-signal transducers and activators of transcription pathway[49]. The presence of pro-apoptotic signaling and the relationship of RIA to known immune pathways in *A. aegypti* is an attractive topic of future study.

In addition to viral particles, ZIKV and DENV infections produce secreted viral nonstructural protein 1, which is present in host serum or culture supernatant[50,51]. Soluble nonstructural protein 1 in the blood meal has been found to increase virus acquisition by mosquitoes, suggesting the protein acts on the

mosquito midgut to enhance viral uptake or antagonize the mosquito antiviral response[51,52]. As we used infected cell culture supernatant rather than purified virus in our mosquito infections, the potential for nonstructural protein 1 and other secreted virus or host factors to impact the RIA response merits further investigation.

In summary, we found that *Ae. aegypti* midgut epithelial cells undergo RIA within 2 hours of exposure to DENV-2 and ZIKV, and that this response is likely caspase mediated. We noted that inhibiting RIA via hAAT treatment increased virus proliferation in the midgut. We propose that RIA beginning 1–4 hpi is a mosquito response to virus exposure distinct from blood meal- or

virus-induced cell death that occurs at later times post infection. The increase in infection severity upon suppression of RIA suggests this phenomenon plays a significant role in mediating the midgut infection barrier in vector mosquitoes (Fig. 5).

## Methods

**Statistics and reproducibility**. Statistical analyses were performed using RStudio Desktop, Open Source Edition and R version 4.0.0[53]. All experiments analyzed represent data from three independent replicates, which were pooled for analysis. Supplementary figs. 2 and 3 represent only two independent replicates and therefore do not include reported statistics. Sample size varies by experiment and is reported in figure legends. Each replicate used mosquitoes from a different hatching, which were fed, dissected, and processed completely separately from other replicates, using common reagents. Data was tested for normality using Shapiro–Wilk tests. For TUNEL-positive cell count experiments, statistical significance was analyzed via Kruskal–Wallis omnibus test with Mann–Whitney pairwise post hoc test. Infection intensity data from plaque assay and RT-qPCR results were analyzed by Mann–Whitney test. Differences in infection prevalence from plaque assay data were analyzed by a chi-squared test. All hypothesis testing was performed at $\alpha = 0.05$.

**Virus culture**. Vero C1008 (clone E6, ATCC) African green monkey kidney cells were grown in complete Dulbecco's Modified Eagle Medium (10% FBS) until 90% confluent, then transferred to reduced Dulbecco's Modified Eagle Medium (DMEM) (3% FBS) for 24 hours before inoculation. Virus was allowed to adsorb for 2 hours at 37 °C, after which media was replaced with non-infectious reduced DMEM. Flasks were observed every 3 days. The supernatant was harvested when 50% of the cells showed visible signs of flavivirus induced cytopathic effects. Cell debris was spun out of the stock, which was then stabilized with 15% trehalose and stored in liquid nitrogen. Titration of stock virus was done by plaque assay on baby hamster kidney cells (BHK-21 Clone 13, ATCC). Strains used in this study were: DENV-2 New Guinea C strain (passage 3, $10^6$ PFU/mL; BEI resources) and ZIKV Puerto Rico PRVABC59 (passage 3, $10^6$ PFU/mL; BEI resources).

**Mosquito rearing**. All mosquito strains (MOYO-R, MOYO-S, and Orlando) were housed in 27 °C growth chambers at 80% humidity on a 12/12 light/dark cycle. Eggs were hatched in warmed deionized (DI) water. Larvae were grown at a density of 400 per 18" × 12.5" larval tray and were fed a 50:50 mix of beef liver powder and ground TetraMin® fish flakes. Pupae were removed daily using plastic transfer pipets. Adults were maintained on a 10% sucrose solution.

**In vivo infection**. Live mosquito infection was performed under Biosafety Level (BSL) 2+ practice in an Arthropod Containment Level 3 facility. Virus-infected blood meals were a 2:2:1 mixture of O+ human RBCs, $10^6$ PFU/mL virus stock, and heat-inactivated human serum. For naive blood meals, the virus stock was replaced by Vero E6 culture supernatant with 15% trehalose. Mosquitoes were fasted on DI water for 12–24 hours prior to blood-feeding and blood fed using a water-jacketed membrane feeder maintained at 37 °C for 30 minutes. For experiments including MOYO-R or MOYO-S mosquitoes, pork sausage casing was used as the membrane, whereas parafilm was used for ORL mosquitoes. After blood feeding, mosquitoes were cold anesthetized (−20 °C) and those that had not blood fed were killed and discarded. At the harvest timepoint, mosquitoes were cold anaesthetized, rinsed 1× in 50% EtOH, and 2× in PBS, and dissected in PBS. Dissected midguts were immediately transferred to 4% paraformaldehyde in fixation buffer (10 mM EGTA, 1 mM MgSO$_4$, 100 mM PIPES (pH 6.9), 75 mM sucrose, 0.1% Triton X-100, in H$_2$O).

**hAAT treatment**. Clinical grade hAAT (Grifol's therapeutics, Catalog # NDC 13533-702-11) was dissolved in autoclaved DI water and added to the blood meal or inoculum at a concentration of 10 mg/mL. The volume of drug was subtracted from the volume of serum added. Negative controls had an equivalent volume of serum or stock replaced by autoclaved DI water.

**Ex vivo infection**. Ex vivo infections were performed in a BSL-2 facility. Non-blood-fed female mosquitoes were dissected in plain DMEM and 10 midguts per infection were pooled, then transferred to virus stock (or Vero E6 cell culture supernatant with 15% trehalose for negative controls) diluted 2:3 into plain DMEM to match the concentration of virus in blood meals for the in vivo infection. Midguts were incubated at room temperature on a rocker for 1–2 hours as the experiment required.

**Midgut plaque assay**. Midguts were dissected, submerged in 150 μL reduced serum DMEM (3% FBS) and stored at −80 °C. After thawing, midguts were homogenized using a Next Advance Bullet Blender 24, and debris was pelleted. Supernatant was serially diluted 10-fold out to $10^3$ in reduced serum DMEM and applied to baby hamster kidney cells (BHK-21 Clone 13, ATCC) for 1 hour. Inoculum was replaced with 0.8% methylcellulose in reduced serum DMEM and

cells were incubated for 4 days before crystal violet stain was applied. Plaques were counted manually.

**Viral genome quantification**. RNA was extracted from individual whole mosquitoes stored in 150 μL Trizol® reagent (Thermo Fisher Scientific Catalog # 15596018) using a Zymo Quick-RNA™ MiniPrep kit (Catalog # R1055). cDNA was prepared from 1000 ng of RNA using a high capacity cDNA reverse transcription kit (Applied Biosystems Ref# 4368814) and a thermocycling program of 25 °C for 10 minutes, 37 °C for 2 hours, and 85 °C for 5 min, followed by a 4 °C indefinite hold. qPCR was performed in duplicate for each target using an Applied Biosystems 7500fast Real-Time PCR System. Primer information can be found in Supplementary Table 2. Primers were designed in house using Primer 3[54], except AeDNR1 primers which were sourced from a previous study[18]. Amplicons were sequenced to confirm specificity. The thermocycler program used was 50 °C for 20 seconds, 95 °C for 10 min; followed by 40 cycles of 95 °C for 15 seconds and 60 °C for 1 min; followed by a melting curve stage (15 seconds at 95 °C, 1 min at 60 °C, 30 seconds at 95 °C, 15 seconds at 60 °C). Delta-CT (ΔCT) values were calculated by subtracting the RPL32 CT value from the viral genome CT value. Relative expression values were calculated as $2^{-\Delta\Delta CT}$, where ΔΔCT was found by subtracting the arithmetic mean ΔCT of the naive blood control across all replicates from each sample ΔCT.

**TUNEL**. Pools of 7–10 midguts were fixed in 4% paraformaldehyde in 5 mM PIPES, pH 7.2 5 mM NaCl, 5 mM MgCl$_2$, 1 mM EGTA (PME buffer) for 30 minutes. They were dehydrated in 25%–50%–75%–100% methanol in a Tris-buffered saline (TBS) gradient with 5 minutes in each step and stored at 4 °C in 100% methanol until processing. Midguts were rehydrated in the reverse gradient. If midguts were from recently blood-fed mosquitoes, they were split open and flattened with a tungsten needle so the blood bolus could be removed. Midguts were then permeabilized in 20 μg/mL proteinase K for 5 minutes at room temperature, then rinsed 2×, washed for 5 minutes 2×, and fixed for 10 minutes in 4% paraformaldehyde in TBS. TUNEL reactions were performed using equilibration buffer and labeling mix from the Millipore FragEL™ DNA Fragmentation Detection Kit, Fluorescent (FITC conjugated) (QIA39-1EA) used as recommended by the manufacturer with 15 U Calf Thymus TdT (Millipore PF060). Midguts were whole mounted for observation on a Zeiss Axioplan 2 imaging microscope, and representative images were taken using a Keyence BZ-X700. TUNEL-positive cells were manually counted.

**Pan-caspase activation assay**. At 1-hour post infection, inoculum was removed from ex vivo-infected guts and replaced with 300 μL of plain DMEM. Red-VAD-FMK (MBL catalog # JM-K190) was added to the media at the concentration recommended by the manufacturer and tissues were incubated for an additional hour at room temperature. Tissues were washed in the provided wash buffer and resuspended in wash buffer for whole-mount fluorescence microscopy. A Zeiss Axioplan 2 imaging microscope was used to visualize the assay, and imaging was completed within 20 minutes of the end of the incubation period.

**Reporting summary**. Further information on research design is available in the Nature Research Reporting Summary linked to this article.

## Data availability
The authors confirm that the data supporting the findings of this study are available in the Supplementary Data 1 file included in the manuscript. No data types with mandatory deposition were used in this study.

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

## Acknowledgements

We thank Dr. Sihong Song for discussion on hAAT and providing us clinical grade hAAT and the United States Department of Agriculture's Center for Medical, Agricultural and Veterinary Entomology for the *A. aegypti* Orlando strain. We thank Dr. David W. Severson and his group for hypothesis generation and collaboration on previous work in this vein as well as for sharing the MOYO strains, and Alyssa Cornista and Ashley Gomez for technical support. This research was supported in part by the United States Centers for Disease Control (CDC) Grant 1U01CK000510-03: Southeastern Regional Center of Excellence in Vector-Borne Diseases: The Gateway Program. The CDC had no role in the design of the study, the collection, analysis, and interpretation of data, or in writing the manuscript. Support was also provided by the University of Florida Emerging Pathogens Institute and the University of Florida Preeminence Initiative (RRD) and NIH GM106174 (LZ). The following reagents were obtained through BEI Resources, NIAID, NIH: Purified Zika Virus, PRVABC59, NR-50684 and Dengue Virus Type 2, New Guinea C, NR-51427. Workflow figures and summary figure were created with Biorender.com.

## Author contributions

The bulk of technical work and experimental design was done by J.B.A. H.G.C. designed the statistical analysis plan and aided J.B.A. in analyzing data. Protocols for hAAT co-feeding and plaque assay were developed by S.K., and S.K. performed or assisted in mosquito feeds and dissections for preliminary data as well as the first replicate of experiments reported in Figs. 1 and 2. L.Z. and R.R.D. provided guidance on project design and writing. LZ performed the bioinformatics analysis of pro-apoptotic genes in the updated AaegL5 genome and helped to optimize the TUNEL protocol. The

manuscript was written by J.B.A. and H.G.C., with R.R.D. and L.Z. contributing significantly to editing and R.R.D. designing most figures.

## Competing interests
The authors declare no competing interests.
