## [Peer Review File · Communications Biology]

Reviewers' comments:

Reviewer #1 (Remarks to the Author):

In this study, J Ayres and co-authors investigate the role of apoptosis in the mosquito defence response against pathogenic flaviviruses ZIKV and DENV. These mosquito-borne flaviviruses cause re-occurring outbreaks and pose a constant threat to the public health in many countries around the globe. To date, the vast majority of the studies on flavivirus-host interactions have been focused on antiviral response in mammals, with flavivirus-mosquito interactions attracting significantly less research attention. In this manuscript authors demonstrate for the first time that infection with ZIKV and DENV induces profound apoptosis in midguts of mosquito within first 1-4h after exposure to the infectious blood meal. Authors also demonstrate that the ability to develop extensive apoptosis in midgut correlates with resistance of the certain mosquito strain to DENV infection, whereas a susceptible mosquito strain shows significantly less apoptosis induction upon infection. Ayres et al. also show that inhibition of apoptosis increases virus replication in mosquito midguts, providing an evidence for the antiviral function of the observed apoptosis induction. Based on their results authors conclude that rapid induction of apoptosis in midguts plays an important role in mediating the midgut infection barrier in vector mosquitoes. This study is novel as apoptosis induction in mosquito midguts at very early time points after infection has not been previously reported. It is well-executed, and conclusions are justified. I read this manuscript with interest and believe it will also be of interest for a wider audience of virologists.

I have the following major concern:

1. Data in figures 1e, 2e and 3e appear not to be normally distributed and therefore cannot be analysed by ANOVA. Authors should use Mann-Whitney test for pairwise comparisons of Kruskal-Wallis test for multiple comparisons. As mean values are not the appropriate descriptive statistics values for data without normal distribution, authors should replace them with median value. It is also unclear what authors wanted to show with SEMs on the graphs. Authors may wish to consult a statistician to ensure appropriate data presentation and analysis.

Minor suggestion:

1. Can authors provide a supplementary figure to support a statement that DMSO alone inhibited apoptosis in mosquito midguts (lines 135-138).

Reviewer #2 (Remarks to the Author):

The manuscript describes the phenomenon of rapid induction of apoptosis (RIA) in mosquito midguts soon after exposure to Flavivirus infection. The phenomenon of RIA has been described in mammalian and insect cells and in *Drosophila melanogaster* in response to infection with Flock house virus (an insect specific virus). This is the first demonstration of the RIA in response to medically important mosquito transmitted viruses; by demonstrating the phenomenon in response to the two pathogenic Flaviviruses dengue and Zika. As the authors discuss, apoptosis has been described in the context of arbovirus infection and has been associated with both pro- and antiviral effects on different arboviruses. However, previous investigations have not assessed early time points (within four hours) after an infectious blood meal; missing the onset of RIA. The authors demonstrate through the use of refractory and susceptible mosquito strains and incorporation of an inhibitor of apoptosis that the RIA phenomenon has significant and substantial effects on virus proliferation and may be a mechanism involved in development of infection resistance in mosquito lines.

The findings are novel to the biology of these mosquito-borne Flaviviruses and their mosquito vectors and have public health significance. The findings substantially impact current understanding of processes occurring at early stages of infection of mosquitoes. This is important and timely as strategies are currently being developed to try to develop refractory mosquito lines that could be used in interventions to reduce the transmission of these viruses. The manuscript presents appropriate and elegantly designed experiments to characterize RIA. Traditional membrane feeding of mosquitoes with blood meals in which virus is present or absent was applied; as well as a highly novel experimental protocol testing the effect of virus in mosquito midguts ex-vivo, in which dissected midguts were placed into virus culture supernatant. RIA was observed in

these midguts. A novel hypothesis is presented to describe the role of RIA in mosquito infection and virus proliferation. Controls were applied appropriately. The manuscript is well written.

Specific comments

Line 65-66. The authors state that "One line of evidence is that viral genes with anti apoptotic function are crucial for infectivity of insects in several families of virus, including ZIKV". The authors could add that viral subgenomic RNA structures have recently been implicated in the regulation of apoptosis in mosquito hosts by ZIKV (Slonchak et al. Nat. Comm. 2020. 11:2205).

Line 81. Correct "focused on at +24 hours"

Line 130 it would be helpful if the titre of the DENV-2 (Jamaica 1409 strain) that was fed was stated.

Line 140. State 'human' before Alpha-1 (to be consistent with the abbreviation) instead of on line 141.

Line 143. Grammar. "the acute infection response"

Lines 144-145. Choose a more specific term than "immune stimulation"

Line 145-146. Was induction of hAAT specifically tested in this study?

Line 148-149. Change to "supplementation of the infective blood meal with 10 mg/mL of clinical grade hAAT (serum level of hAAT in healthy subjects is around 1.5-3.5 mg/mL) suppressed RIA.."

Line 151-152 and elsewhere. P numbers below 0.001 need only be stated as $p < 0.001$.

Line 158 Cite Fig 4f for the ZIKV result.

Line 166 State the test used to measure correlation between hAAT and mortality in the brackets.

Line 219. Change "via hAAT" to "via hAAT treatment" or "using hAAT" or something similar.

Line 273. Confirm feeders were maintained at 39°C.

Line 284. The statement says that either serum or virus was replaced with the drug. I am not sure why a consistent approach was not used. Can you indicate which experiments either approach was applied.

Lines 315-316. State the calculation of delta CT after stating the qPCR cycling conditions.

Figure 5. Showing 2 and 24 hr time points for both a and b may be helpful (if space allows). A shaded cell labelled ISC is prominently shown in all panels, however there is no definition for ISC (intestinal stem cell?) or any reference to ISC (or roles for different epithelial cell types) in the manuscript. Suggest you either make reference to this cell type in the manuscript if it is relevant to the hypothesis or delete the ISC label (and possibly the cells in the cartoons) if these cells are not relevant to the hypothesis.

Extended data figure 1. Two graphs are shown essentially of the same data, but with panel b showing early time points on more restricted y axis (presumably to accentuate differences). I found this confusing at first as there is no description that panel B duplicates data in panel A. Further, duplication of data is to be avoided. I suggest another approach is used to accentuate low and high differences on a single graph (possibly a split y axis or log scale). Statistical annotation would also be useful here showing the significant comparisons between the ZIKV and naïve groups at each time point (as done for the other graphs).

Line 380. Comma not needed after hAAT

Figure 4f – provided data. Replicate 3 data seems to have yielded substantially higher numbers of plaque forming units for all treatments when compared to the first two replicates. Is there a reason for this? Given that the increase seems to be proportional across treatments this likely mitigates bias for any one treatment.

Reviewer #3 (Remarks to the Author):

Ayers J. et al. showed that the rapid induction of apoptosis after feeding of infected blood meal leads to the reduction of flavivirus replication in midgut. Although the importance of apoptosis in midgut during virus replication has been already shown, this study focused on the very early event in midgut. They first demonstrated the rapid induction of apoptosis in vivo and ex vivo. Then, they examined if the susceptibility to viruses affects the rapid induction of apoptosis. In addition, they confirmed that inhibition of caspase-3 activation leads to the reduction of TUNEL positive cells after infected blood meal feeding. Given the many question marks around the relationships between blood feeding and virus infection, the topic of the manuscript is very interesting and highly

relevant. However, the data are descriptive and does not show any mechanism underlying the rapid induction of apoptosis.

I have the following questions and concerns:

Major comments

1: In the infection model, they use the supernatant of virus-infected cells. The supernatant in virus-infected cells potentially contains a lot of molecules to stimulate antiviral genes, which may affect the induction of apoptosis. To avoid this possibility, for example, they need to use the purified virions for the assays.

2: This manuscript suggests the hypothesis but does not show any underlying mechanism of the rapid induction of apoptosis. Since they describe some key genes during apoptosis, at least they should check the expression levels of these factors.

3: In figure 4, they described the inhibition of the rapid induction of apoptosis results in viral replication by using hAAT. If the apoptosis induced at 24+ hours post infection is also inhibited, it is difficult to distinguish which time points of inhibition of apoptosis leads to these data. How authors would respond to that kind of criticism?

Minor comments

4: In the text, they used both "Ae aegypti" and "Ae. aegypti". Please unify to "Ae. aegypti".

5: Line 117-118, they first show the Fig. 2e, then explain Fig. 2b-d. I think the order of figures and explanation in the text should be the same. (I mean, please first explain Fig. 2b-d, and then explain Fig. 2e in the text.)

6: In the methods, they need to show the bio-safety levels when they performed mosquito works and ex vivo works.

Author responses to reviewer comments are given in **blue**.

Reviewers' comments:

Reviewer #1 (Remarks to the Author):

In this study, J Ayres and co-authors investigate the role of apoptosis in the mosquito defence response against pathogenic flaviviruses ZIKV and DENV. These mosquito-borne flaviviruses cause re-occurring outbreaks and pose a constant threat to the public health in many countries around the globe. To date, the vast majority of the studies on flavivirus-host interactions have been focused on antiviral response in mammals, with flavivirus-mosquito interactions attracting significantly less research attention. In this manuscript authors demonstrate for the first time that infection with ZIKV and DENV induces profound apoptosis in midguts of mosquito within first 1-4h after exposure to the infectious blood meal. Authors also demonstrate that the ability to develop extensive apoptosis in midgut correlates with resistance of the certain mosquito strain to DENV infection, whereas a susceptible mosquito strain shows significantly less apoptosis induction upon infection. Ayres et al. also show that inhibition of apoptosis increases virus replication in mosquito midguts, providing an evidence for the antiviral function of the observed apoptosis induction. Based on their results authors conclude that rapid induction of apoptosis in midguts plays an important role in mediating the midgut infection barrier in vector mosquitoes. This study is novel as apoptosis induction in mosquito midguts at very early time points after infection has not been previously reported. It is well-executed, and conclusions are justified. I read this manuscript with interest and believe it will also be of interest for a wider audience of virologists.

I have the following major concern:

1. Data in figures 1e, 2e and 3e appear not to be normally distributed and therefore cannot be analysed by ANOVA. Authors should use Mann-Whitney test for pairwise comparisons of Kruskal-Wallis test for multiple comparisons. As mean values are not the appropriate descriptive statistics values for data without normal distribution, authors should replace them with median value. It is also unclear what authors wanted to show with SEMs on the graphs. Authors may wish to consult a statistician to ensure appropriate data presentation and analysis.

Response 1. **First, we sincerely appreciate the reviewer's overall positive assessment of the novelty of our work. Furthermore, we thank the reviewer for the statistical critique and have re-run our data using a Kruskal-Wallis omnibus test with Mann-Whitney pair-wise comparisons post-hoc, as suggested. The authors were under the assumption that ANOVAs are generally robust against non-normality but not heteroscedasticity, and as such, a Welch's one-way ANOVA was previously used. We have updated the methods section and figure captions accordingly and changed the descriptive statistic shown on the graphs to median values. It is worth noting that the conclusions of the analysis remain the same.**

Minor suggestion:

1. Can authors provide a supplementary figure to support a statement that DMSO alone inhibited apoptosis in mosquito midguts (lines 135-138).

Response 2. Extended data figure 2, which shows the reduction in TUNEL positive cells in DMSO vehicle controls has been added.

Reviewer #2 (Remarks to the Author):

The manuscript describes the phenomenon of rapid induction of apoptosis (RIA) in mosquito midguts soon after exposure to Flavivirus infection. The phenomenon of RIA has been described in mammalian and insect cells and in *Drosophila melanogaster* in response to infection with Flock house virus (an insect specific virus). This is the first demonstration of the RIA in response to medically important mosquito transmitted viruses; by demonstrating the phenomenon in response to the two pathogenic Flaviviruses dengue and Zika. As the authors discuss, apoptosis has been described in the context of arbovirus infection and has been associated with both pro- and antiviral effects on different arboviruses. However, previous investigations have not assessed early time points (within four hours) after an infectious blood meal; missing the onset of RIA. The authors demonstrate through the use of refractory and susceptible mosquito strains and incorporation of an inhibitor of apoptosis that the RIA phenomenon has significant and substantial effects on virus proliferation and may be a mechanism involved in development of infection resistance in mosquito lines.

The findings are novel to the biology of these mosquito-borne Flaviviruses and their mosquito vectors and have public health significance. The findings substantially impact current understanding of processes occurring at early stages of infection of mosquitoes. This is important and timely as strategies are currently being developed to try to develop refractory mosquito lines that could be used in interventions to reduce the transmission of these viruses. The manuscript presents appropriate and elegantly designed experiments to characterize RIA. Traditional membrane feeding of mosquitoes with blood meals in which virus is present or absent was applied; as well as a highly novel experimental protocol testing the effect of virus in mosquito midguts ex-vivo, in which dissected midguts were placed into virus culture supernatant. RIA was observed in these midguts. A novel hypothesis is presented to describe the role of RIA in mosquito infection and virus proliferation. Controls were applied appropriately. The manuscript is well written.

Response 3. We thank the reviewer for recognizing the novelty and potentially utility of the work in informing the design of public health interventions.

Specific comments

Line 65-66. The authors state that “One line of evidence is that viral genes with anti-apoptotic function are crucial for infectivity of insects in several families of virus, including ZIKV”. The authors could add that viral subgenomic RNA structures have recently been implicated in the regulation of apoptosis in mosquito hosts by ZIKV (Slonchak et al. Nat. Comm. 2020. 11:2205).

Response 4. While we had referenced this publication, the significance of the reference was not well explained. This point has been added to the text. We thank the reviewer for their comments and corrections.

Line 81. Correct “focused on at +24 hours”

Response 5. Corrected.

Line 130 it would be helpful if the titre of the DENV-2 (Jamaica 1409 strain) that was fed was stated.

Response 6. In the previous study on MOYO strain vector competence cited here, a titer of the infectious blood meal was not performed. The authors fed a cell suspension (60%) of C6/36 cells, which had been infected at an MOI of 0.1 14 days prior. In the DENV-2 infection of MOYO-R and MOYO-S performed in our lab we maintained a consistent 10⁶ PFU/mL for *in vitro* infections (Lines 276-279 in the original submission). We have edited the text as follows:

“When both MOYO-R and MOYO-S mosquitoes were fed with a DENV-2-infected blood meal (10⁶ PFU/mL), there was a 1.8-fold higher mean number of TUNEL-positive epithelial cells in MOYO-R mosquitoes as compared to MOYO-S mosquitoes ($p = 0.00212$) (**Fig. 3**).” (Lines 134-137)

Line 140. State ‘human’ before Alpha-1 (to be consistent with the abbreviation) instead of on line 141.

Response 7. Revised as recommended.

Line 143. Grammar. “the acute infection response.”

Response 8. Corrected.

Lines 144-145. Choose a more specific term than “immune stimulation”

Response 9. Corrected as follows: “hAAT is also involved in the acute infection response as up to a 4-fold increase in hAAT serum concentration was found following stimulation of innate immune cells in donor blood...”

Line 145-146. Was induction of hAAT specifically tested in this study?

Response 10. In the previous study cited here, alpha-1 antitrypsin was directly measured by ELISA in plasma from control patients and dengue patients, and no significant difference was observed.

Line 148-149. Change to “supplementation of the infective blood meal with 10 mg/mL of clinical grade hAAT (serum level of hAAT in healthy subjects is around 1.5-3.5 mg/mL) suppressed RIA.”

Response 11. Edited as recommended.

Line 151-152 and elsewhere. P numbers below 0.001 need only be stated as $p < 0.001$.

Response 12. *Communications Biology* asks in the author materials that exact P values be given wherever possible, so we would prefer to continue reporting exact p values in the text.

Line 158 Cite Fig 4f for the ZIKV result.

Response 13. Changed as noted.

Line 166 State the test used to measure correlation between hAAT and mortality in the brackets.

Response 14. Changed as follows: "There was no observable correlation between mortality and hAAT supplementation (for ZIKV $p = 0.183$, for DENV-2 $p = 0.637$, by Chi-squared test) (**Supplementary Table 1**). "

Line 219. Change "via hAAT" to "via hAAT treatment" or "using hAAT" or something similar.

Response 15. Changed as noted.

Line 273. Confirm feeders were maintained at 39°C.

Response 16. To ensure ~37°C temperature maintenance across all feeders (due to tubing length from the circulation water bath outlet) the circulation water bath is set to 39°C. To avoid confusion, we have corrected the temperature to 37°C in the revised text.

Line 284. The statement says that either serum or virus was replaced with the drug. I am not sure why a consistent approach was not used. Can you indicate which experiments either approach was applied.

Response 17. The volume of drug was always subtracted from the serum content of the blood meal. This statement has been corrected.

Lines 315-316. State the calculation of delta CT after stating the qPCR cycling conditions.

Response 18. Changed as noted.

Figure 5. Showing 2 and 24 hr time points for both a and b may be helpful (if space allows). A shaded cell labelled ISC is prominently shown in all panels, however there is no definition for ISC (intestinal stem cell?) or any reference to ISC (or roles for different epithelial cell types) in the manuscript. Suggest you either make reference to this cell type in the manuscript if it is relevant to the hypothesis or delete the ISC label (and possibly the cells in the cartoons) if these cells are not relevant to the hypothesis.

Response 19. We thank the reviewer for identifying this issue. To increase clarity, we have elected to delete the reference to the ISCs in Figure 5.

Extended data figure 1. Two graphs are shown essentially of the same data, but with panel b showing early time points on more restricted y axis (presumably to accentuate differences). I found this confusing at first as there is no description that panel B duplicates data in panel A. Further, duplication of data is to be avoided. I suggest another approach is used to accentuate low and high differences on a single graph (possibly a split y axis or log scale). Statistical annotation would also be useful here showing the significant comparisons between the ZIKV and naïve groups at each time point (as done for the other graphs).

Response 20. We apologize for the oversight in ensuring clarity in what was presented. Panel b is essentially an “inset” so that a reader can appreciate the variation in the earlier time points (i.e., data points are not necessarily close to baseline) and less apparent differences between treatment and controls (as correctly presumed). The use of a two-sectioned y-axis would allow for all data to be shown in a single image and we have elected to use this approach to convey the results. A third replicate of this experiment was performed to allow robust statistical analysis, and the results of this analysis were added to the figure as recommended.

Line 380. Comma not needed after hAAT.

Response 21. Corrected.

Figure 4f – provided data. Replicate 3 data seems to have yielded substantially higher numbers of plaque forming units for all treatments when compared to the first two replicates. Is there a reason for this? Given that the increase seems to be proportional across treatments this likely mitigates bias for any one treatment.

Response 22. There is no clear reason for the higher titers in replicate 3. The virus stock was constant across replicates and was used as a positive control to confirm that plaque assay detection of virus was consistent. However, the mosquitoes from replicate studies represent separate biological cohorts sampled from the primary mosquito colony, so between-replicate variability in infection rate was not unexpected.

Reviewer #3 (Remarks to the Author):

Ayers J. et al. showed that the rapid induction of apoptosis after feeding of infected blood meal leads to the reduction of flavivirus replication in midgut. Although the importance of apoptosis in midgut during virus replication has been already shown, this study focused on the very early event in midgut. They first demonstrated the rapid induction of apoptosis in vivo and ex vivo. Then, they examined if the susceptibility to viruses affects the rapid induction of apoptosis. In addition, they confirmed that inhibition of caspase-3 activation leads to the reduction of TUNEL positive cells after infected blood meal feeding. Given the many question marks around the relationships between blood feeding and virus infection, the topic of the manuscript is very

interesting and highly relevant. However, the data are descriptive and does not show any mechanism underlying the rapid induction of apoptosis.

Response 23. We thank the reviewer for affirming the relevance and potentially wider interest in the phenomenon captured through this study. The study does indeed describe the first demonstration of RIA at a very early time point, opening avenues to dissect the exact mechanism. We concur with Reviewer 2 that we offer a novel hypothesis to describe the role of RIA in mosquito infection and virus proliferation that we anticipate being able to test in greater detail in subsequent studies.

I have the following questions and concerns:

Major comments

1: In the infection model, they use the supernatant of virus-infected cells. The supernatant in virus-infected cells potentially contains a lot of molecules to stimulate antiviral genes, which may affect the induction of apoptosis. To avoid this possibility, for example, they need to use the purified virions for the assays.

Response 24. We thank the reviewer for their comments. While it is an interesting point that stress signals from the infected mammalian cells used to culture virus may play a role in insect immune response, we hesitate to compare purified virus to culture supernatant due to the damaging nature of density gradient purification, which we expect to reduce virus infectivity in ways that could not be completely accounted for by re-titering in a susceptible cell line. Additionally, secreted viral proteins contained in the blood meal but not be present in the virion itself, such as Zika virus NS1, which have been shown to have significant impact on viral infectivity and mosquito immune response (<https://doi.org/10.1038/s41467-017-02816-2>) and this facet would be lost in an infection with purified virus. In a certain sense, a naturally infected mosquito is feeding on infected culture supernatant when it bites an infected host. Therefore, we feel that while investigating the precise nature of the interaction between the mosquito midgut and the infected blood meal that causes RIA is a worthwhile topic of future study, clearly differentiating between viral and host cell factors would be a complicated process outside of the scope of this manuscript.

2: This manuscript suggests the hypothesis but does not show any underlying mechanism of the rapid induction of apoptosis. Since they describe some key genes during apoptosis, at least they should check the expression levels of these factors.

Response 25. We feel that the mechanism of RIA is an important topic of future study but is largely outside the scope of this manuscript, and that the descriptive outline of a novel phenomenon provided in this manuscript will be of interest to others in the field. However, in response to this criticism, we began to address this question by selecting 3 upstream regulators of caspase activation which have previously been implicated in mosquito innate immunity and analyzing transcript level at 2hpi by rt-qPCR. We chose *IMP* and *mx*, IAP antagonists which previous studies have suggested may be transcriptionally upregulated (as measured by microarray and RNAseq) during the RIA response, as well as *DNR1*, a negative apoptosis regulator also involved in suppressing innate immune signaling through the IMD pathway. Transcript level of *IMP* and *DNR1* do not change in virus-fed mosquitoes. Transcript level of *mx*

is significantly higher in ZIKV-fed than naive blood-fed mosquitoes but does not increase in DENV-2 fed mosquitoes. These new data are included in the revised manuscript as Extended Data Figure 4. We hope to follow up on the potential role of *mx* and its regulation at the transcriptomic and proteomic levels in the future.

3: In figure 4, they described the inhibition of the rapid induction of apoptosis results in viral replication by using hAAT. If the apoptosis induced at 24+ hours post infection is also inhibited, it is difficult to distinguish which time points of inhibition of apoptosis leads to these data. How authors would respond to that kind of criticism?

Response 26. We did not expect hAAT to retain activity throughout the process of bloodmeal digestion, and therefore predicted that its anti-apoptotic effect would be very short-lived. The 24 hours following a blood meal are a period of highly active midgut proteolytic processing (midgut secreted Early Trypsin activity peaks >3 hrs post feeding followed by late chymotrypsin/activity peaking between 18-24 hrs, followed by endo- and exopeptidase activity from peritrophic matrix-associated proteases and finally action on bloodmeal glycoproteins/glycopeptides by midgut apical surface proteases). To support this premise, we have confirmed that hAAT-supplemented and control ZIKV infected mosquitoes showed equivalent levels of DNA fragmentation by 24h post-infection and have added this data to the manuscript in Extended Data Figure 3.

Minor comments

4: In the text, they used both “*Ae aegypti*” and “*Ae. aegypti*”. Please unify to “*Ae. aegypti*”.

Response 27. Thank you for catching this typo. We have checked for consistency to *Ae. aegypti* in the revised text.

5: Line 117-118, they first show the Fig. 2e, then explain Fig. 2b-d. I think the order of figures and explanation in the text should be the same. (I mean, please first explain Fig. 2b-d, and then explain Fig. 2e in the text.)

Response 28. Changed as recommended.

6: In the methods, they need to show the bio-safety levels when they performed mosquito works and ex vivo works.

Response 29. We have edited the methods section as recommended.

REVIEWERS' COMMENTS:

Reviewer #1 (Remarks to the Author):

The authors addressed my comments in full. I don't have any further concerns regarding the manuscript.

Reviewer #2 (Remarks to the Author):

Thank you for addressing my prior comments in this revision. I've identified only the following minor points to be addressed:

Line 70 Error of citation numbering - '16, 14-16'

Line 110-113 The authors appear to coin the term Rapid Induction of Apoptosis (and define the abbreviation RIA) here, but have defined and used the term already in line 85 in the context of another published study. These statements would be most appropriate at the first mention of the term in the body of the manuscript.

Line 188. The statement 'We selected mx, IMP, and DNR1 as upstream regulators of caspase activation...' does not tell the reader for what these proteins were selected for or how they were investigated. Please complete this sentence. Further, please the full names for these proteins and define their abbreviations at their first mention in the body of the manuscript.

Line 192. Change "transcript level of mx was significantly increase" to "transcript levels of mx were significantly increased"

Line 196. Change 'suggested' to 'indicated'

Line 203. It would be useful to state which enhancer or regulatory regions are being referred to in this sentence.

Line 206. Suggest merging this paragraph with the previous paragraph.

Paragraph starting line 217. This topic of this paragraph is the potential for regulation of pro-apoptotic genes by genetic polymorphisms in their regulator regions, however this topic is already discussed in lines 196-204. There seems to be unnecessary duplication. I suggest consolidating writing on this topic together.

Reviewer #3 (Remarks to the Author):

The manuscript has improved and the authors have addressed most of my questions. I have one comment and question.

1: As the authors described in the response to my previous concern, secreted viral proteins contained in the blood meal but not be present in the virion itself, such as Zika virus NS1 may affect RIA. More discussion regarding this point might be better to imply the underlying mechanism of RIA.

2: I may miss, but could authors clarify "a" "b" or "c" in the graphs of figures?

Author responses are in blue.

Reviewer #1 (Remarks to the Author):

The authors addressed my comments in full. I don't have any further concerns regarding the manuscript.

Response 1. We thank the reviewer for their positive assessment of the manuscript.

Reviewer #2 (Remarks to the Author):

Thank you for addressing my prior comments in this revision. I've identified only the following minor points to be addressed:

Response 2. We thank the reviewer for their comments and appreciate their attention to detail in suggesting edits.

Line 70 Error of citation numbering - '16, 14-16'

Response 3. Numbering has been corrected.

Line 110-113 The authors appear to coin the term Rapid Induction of Apoptosis (and define the abbreviation RIA) here, but have defined and used the term already in line 85 in the context of another published study. These statements would be most appropriate at the first mention of the term in the body of the manuscript.

Response 4. We have moved the information on the definition of RIA from line 110 to immediately after the first mention of the phenomenon on line 85.

Line 188. The statement 'We selected mx, IMP, and DNR1 as upstream regulators of caspase activation...' does not tell the reader for what these proteins were selected for or how they were investigated. Please complete this sentence. Further, please the full names for these proteins and define their abbreviations at their first mention in the body of the manuscript.

Response 5. We have clarified our interest in these genes and stated we analyzed their transcript level in the first sentence of this paragraph, as well as added the full names at their first mention.

Line 192. Change "transcript level of mx was significantly increase" to "transcript levels of mx were significantly increased"

Response 6. Corrected.

Line 196. Change 'suggested' to 'indicated'

Response 7. Wording has been changed as recommended.

Line 203. It would be useful to state which enhancer or regulatory regions are being referred to in this sentence.

Response 8. Unfortunately, the study cited here only reported global presence of SNPs in regions putatively identified as regulatory sequences by formaldehyde-assisted isolation of regulatory elements sequencing and did not comment on the specific genes or gene families affected.

Line 206. Suggest merging this paragraph with the previous paragraph.

Response 9. Paragraphs have been merged.

Paragraph starting line 217. This topic of this paragraph is the potential for regulation of pro-apoptotic genes by genetic polymorphisms in their regulator regions, however this topic is already discussed in lines 196-204. There seems to be unnecessary duplication. I suggest consolidating writing on this topic together.

Response 10. The discussion of regulatory regions has been condensed to a single paragraph.

Reviewer #3 (Remarks to the Author):

The manuscript has improved and the authors have addressed most of my questions. I have one comment and question.

1: As the authors described in the response to my previous concern, secreted viral proteins contained in the blood meal but not be present in the virion itself, such as Zika virus NS1 may affect RIA. More discussion regarding this point might be better to imply the underlying mechanism of RIA.

Response 11. We thank the reviewer for their comments and positive assessment of the manuscript. We have added a paragraph discussing the contribution of soluble NS1 to flavivirus infection to the discussion section.

2: I may miss, but could authors clarify "a" "b" or "c" in the graphs of figures?

Response 12. We appreciate the reviewer noticing this error. Panel lettering has been clarified in the legend of supplementary figure 4.